# Climate-related factors cause changes in the diversity of fish and invertebrates in subtropical coast of the Gulf of Mexico

Masami Fujiwara [1]*, Fernando Martinez-Andrade [2], R.J.David Wells [3], Mark Fisher[4], Michaela Pawluk [1] & Mariah C. Livernois [3]

Climate change impacts physical and chemical properties of the oceans, and these changes affect the ecology of marine organisms. One important ecological consequence of climate change is the distribution shift of marine species toward higher latitudes. Here, the prevalence of nearly 150 species of fish and invertebrates were investigated to find changes in their distributions over 35 years along a subtropical coast within the Gulf of Mexico. Our results show that 90 species increased their occupancy probability, while 33 decreased (remaining species neither increase or decrease), and the ranges of many species expanded. Using rarefaction analysis, which allows for the estimation of species diversity, we show that species diversity has increased across the coast of Texas. Climate-mediated environmental variables are related to the changes in the occupancy probability, suggesting the expansion of tropical species into the region is increasing diversity.

---

[1] Department of Wildlife and Fisheries Sciences, Texas A&M University, College Station, TX 77843-2258, USA. [2] Coastal Fisheries Division, Texas Parks and Wildlife Department, 6300 Ocean Dr., NRC Bldg. Suite 2500, Corpus Christi, TX 78412-5845, USA. [3] Department of Marine Biology, Texas A&M University at Galveston, Galveston, TX 77553, USA. [4] Coastal Fisheries Division, Texas Parks and Wildlife Department, 702 Navigation Circle, Rockport, TX 78382, USA. *email: fujiwara@tamu.edu

Climate change affects the range, distribution, and species composition of marine organisms. For example, the ranges of fish have shifted northward in the North Sea[1], the assemblages of marine organisms have changed to include more warm-water species in the northeast United States[2,3] and the North Sea[4], the distribution of organisms has changed in the California Current[5], and warm-water fish have expanded their ranges in the Mediterranean Sea[6]. The species composition in the northeast United States has also been shown to correlate strongly with the spring–summer sea surface temperature[2], and the mean preferred temperature of landed fish increased from 1970 to 2006 worldwide[7].

Several underlying mechanisms linking climate change and these ecological effects have been suggested. For example, changes in the range distribution and species composition are thought to result from reduced oxygen supply resulting from increased temperature[8], temperatures exceeding upper heat limits[9,10], and temperatures exceeding the level for maximum growth[11]. Most of these hypothesized mechanisms suggest that temperature affects organisms directly or indirectly. However, coastal marine species are also affected by other factors such as sea-level rise and salinity, both of which are affected by climate change.

Here, we investigate the prevalence of nearly 150 species of fish and invertebrates to demonstrate changes in their distributions over 35 years along a subtropical coast within the Gulf of Mexico.

Fishery-independent bag seine data were collected biweekly by the Coastal Fisheries Division of the Texas Parks and Wildlife Department starting in 1982 (or 1986 in one bay), as a part of the Marine Resource Monitoring Program[12]. This is the first of this type of study in the region, and is unique in including a large number of species monitored over an extended period of time. To determine the spatiotemporal trends of marine species in each of the eight major bay systems along the Texas coast (Fig. 1), we estimated a bay-wide diversity index[13] and an occupancy probability for individual species[14]. Then, the occupancy probability was associated with water temperature, salinity, dissolved oxygen, and sea level.

Our results suggest that diversity has increased across the coast, the majority of species has increased in prevalence, and the ranges of many species expanded. Climate-mediated environmental variables (sea level and temperature) are related to the temporal changes in the occupancy probability. Although increased diversity may have positive effects on ecosystems, such changes in a short term will likely impact the ecology of these systems by introducing new species interactions or altering existing ones.

## Results

**Annual Shannon diversity index of fish and invertebrates.** The Shannon diversity index was calculated in each major bay system

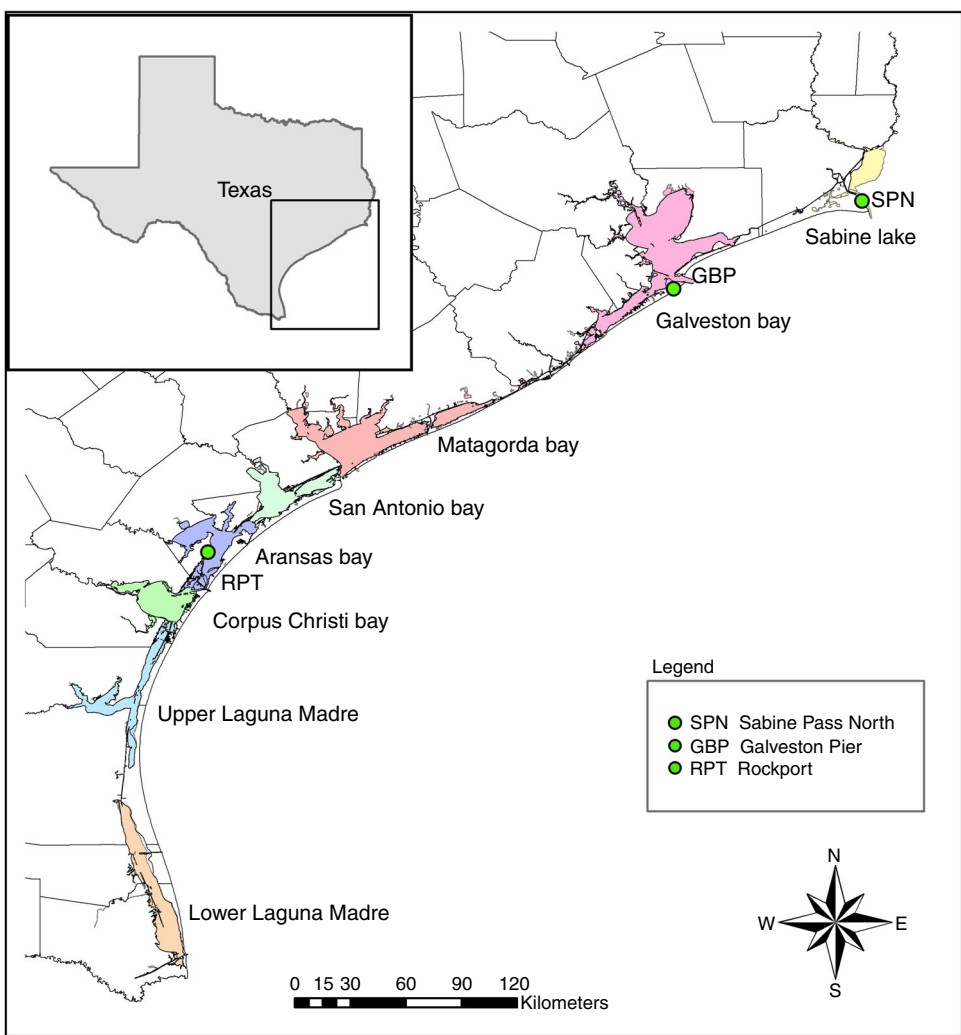

**Fig. 1** Location of bays used in this study. Green circles indicate the locations of sea-level measurements: SPN Sabine Pass North, GBP Galveston Bay Pier, RPT Rockport

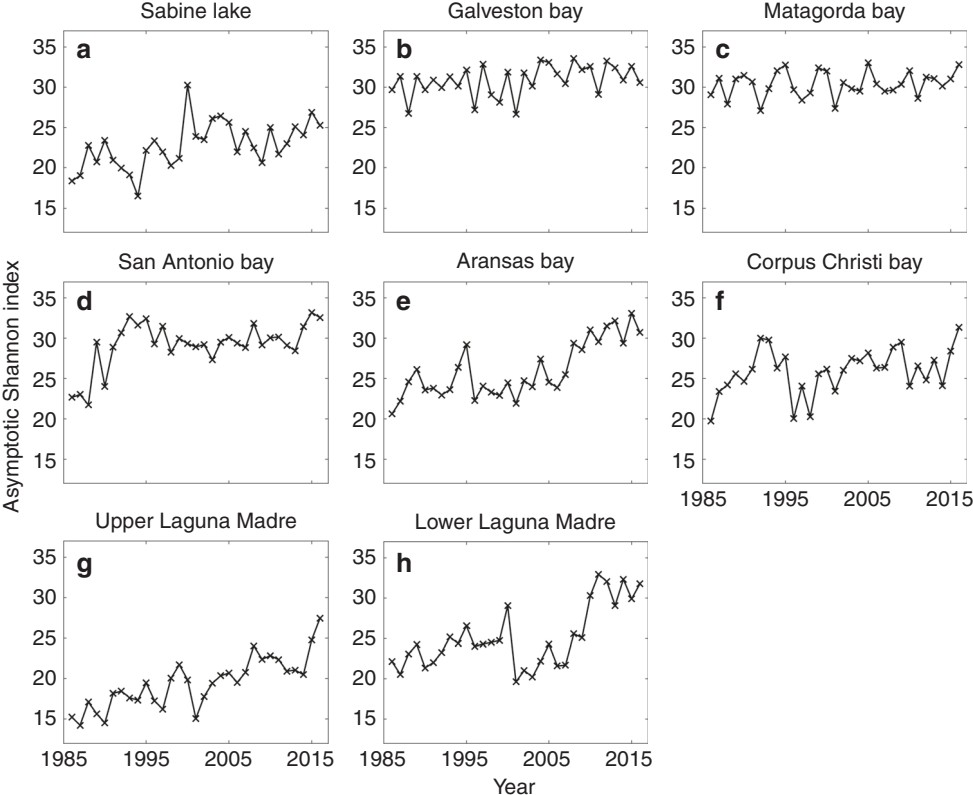

**Fig. 2** Estimated Shannon diversity index of fish species estimated for eight major bays by using a sample-based rarefaction analysis. **a** Sabine Lake, **b** Galveston Bay, **c** Matagorda Bay, **d** San Antonio Bay, **e** Aransas Bay, **f** Corpus Christi Bay, **g** Upper Laguna Madre, and **h** Lower Laguna Madre. The index is shown for different years. The exact sample size varies among locations and years, and it is presented in Supplementary Data 1. An increasing trend in index was significant for Sabine Lake (Holm–Bonferroni adjusted p-value = 0.000), San Antonio Bay (Holm–Bonferroni adjusted p-value = 0.020), Aransas Bay (Holm–Bonferroni adjusted p-value = 0.000), Upper Laguna Madre (Holm–Bonferroni adjusted p-value = 0.000), and Lower Laguna Madre (Holm–Bonferroni adjusted p-value = 0.000) based on a parametric bootstrap method (n = 1000). An increasing trend was not significant in other bays: Galveston Bay (Holm–Bonferroni adjusted p-value = 1.000), Matagorda Bay (Holm–Bonferroni adjusted p-value = 1.000), and Corpus Christi Bay (Holm–Bonferroni adjusted p-value = 1.000). None of the bays showed significantly declining trends. See Supplementary Data 2

to measure species richness while accounting for the evenness of species present, and it was estimated by using rarefaction analysis[13] to account for variability in sample size. The diversity index for fish increased significantly in five of the nine bays and did not decline significantly in any of the bays (Fig. 2). Similarly, the diversity index for invertebrates increased significantly in four bays and did not decline in any (Fig. 3). The results suggest that species diversity is generally increasing in the bays along the Texas coast.

**Monthly occupancy probability of fish and invertebrates.** While the diversity index analysis indicates changes in the overall diversity in the bays, the occupancy probability analysis measures species-specific spatiotemporal trends in prevalence. The analysis was applied to nearly 150 species of fish and invertebrates, and occupancy probability for 87 fish and 43 invertebrate species was estimated (Supplementary Data 4 and 5). Among the 130 species, 90 of them (66% of fishes and 77% of invertebrates) increased their occupancy probability over time (Fig. 4a), and 33 species (30% of fishes and 16% of invertebrates) decreased their probability (Fig. 4b). A total of 27 species (16% of fishes and 30% of invertebrates) expanded their range, and five species (6% of fishes and 2% of invertebrates) contracted their range (Fig. 4c, d). Our results showed that the proportion of declining fish species (26 out of 87 species) was significantly greater than that of declining

invertebrate species (seven out of 43 species; single-tail binomial test, p-value = 0.041).

We further investigated the associations between trends in the occupancy probability of fishes and their maximum preferred/ observed temperature and geographic distribution (southern limit, northern limit, and distribution range). Compared with species that increased in occupancy probability, those species with declining occupancy probabilities had significantly lower-latitude southern range limits (single-tail Mann–Whitney U test, rank sum = 1176, n = [52, 24], Holm–Bonferroni adjusted p-value = 0.004), significantly lower maximum preferred/observed temperatures (single-tail Mann–Whitney U test, rank sum = 834, n = [55, 25], Holm–Bonferroni adjusted p-value = 0.062), and significantly narrower latitudinal distribution ranges (single-tail Mann–Whitney U test, rank sum = 1831, n = [54, 24], Holm–Bonferroni adjusted p-value = 0.084). Similarly, species that contracted their range had significantly higher-latitude southern range limits (single-tail Mann–Whitney U test, rank sum = 56.5, n = [14, 4], Holm–Bonferroni adjusted p-value = 0.088) and significantly lower maximum preferred/observed temperatures (single-tail Mann–Whitney U test, rank sum = 59, n = [4, 4], Holm–Bonferroni adjusted p-value = 0.069) compared with those that expanded their range.

To investigate the effects of climate-related environmental variables on changes in occupancy probability, we fit sea level, temperature, salinity, and dissolved oxygen as covariates in the

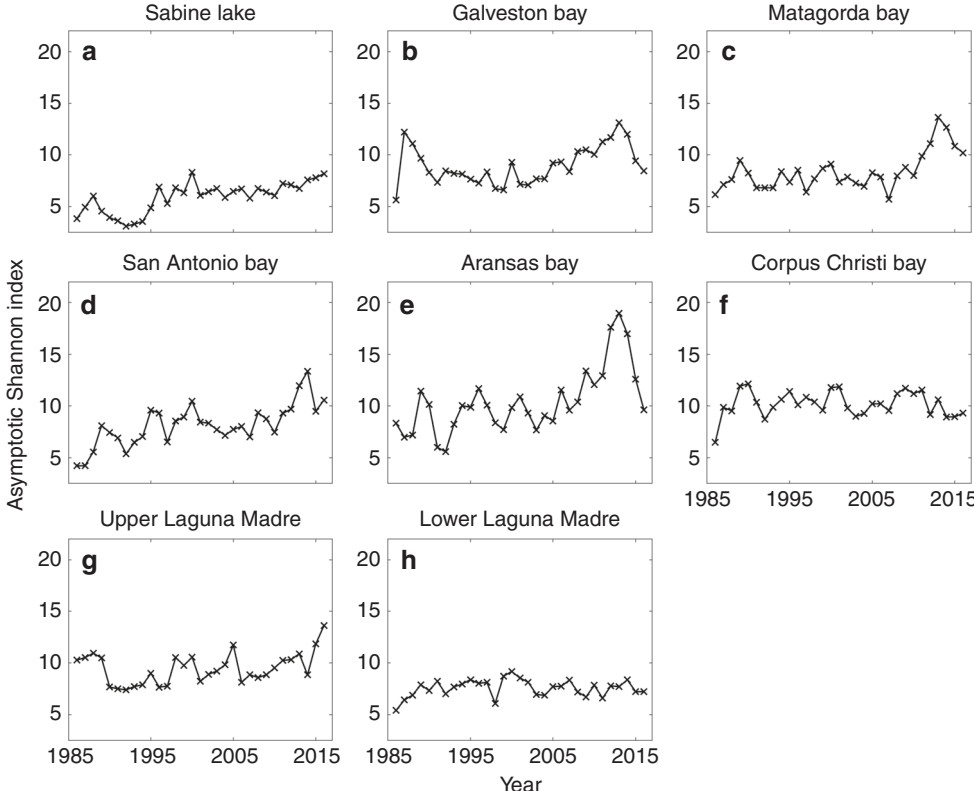

**Fig. 3** Estimated Shannon diversity index of invertebrate species estimated for eight major bays by using a sample-based rarefaction analysis. **a** Sabine Lake, **b** Galveston Bay, **c** Matagorda Bay, **d** San Antonio Bay, **e** Aransas Bay, **f** Corpus Christi Bay, **g** Upper Laguna Madre, and **h** Lower Laguna Madre. The index is shown for different years. The exact sample size varies among locations and years, and it is presented in Supplementary Data 1. An increasing trend in index was significant for Sabine Lake (Holm–Bonferroni adjusted p-value = 0.000), Matagorda Bay (Holm–Bonferroni adjusted p-value = 0.000), San Antonio Bay (Holm–Bonferroni adjusted p-value = 0.000), and Aransas Bay (Holm–Bonferroni adjusted p-value = 0.000) based on a parametric bootstrap method (n = 1000). An increasing trend was not significant in other bays: Galveston Bay (Holm–Bonferroni adjusted p-value = 1.000), Corpus Christi Bay (Holm–Bonferroni adjusted p-value = 1.000), Upper Laguna Madre (Holm–Bonferroni adjusted p-value = 1.000), and lower Laguna Madre (Holm–Bonferroni adjusted p-value = 1.000). None of the bays showed a significantly declining trend. See Supplementary Data 3

occupancy probability model for each species. Variables were included two at a time, with one explaining variability among years and the other explaining the variability among bays and months. This analysis allowed the estimation of occupancy probability for 96 fish and 45 invertebrate species with some of the covariates. For the majority of species, salinity was the best predictor of variability among bays and months (Fig. 5a, b), and change in sea level was the best predictor of variability among years (Fig. 5c, d). However, among the fish species for which variability among years was affected by temperature, a higher proportion exhibited declining trends than species affected by other factors (Fig. 5c; cf ratios between declining and increasing across bars).

## Discussion

The Shannon diversity index for fish and invertebrate significantly increased in most of the bays, and occupancy probably of the majority of fish and invertebrate species increased over time. Our results show that species closer to the southern limit of their historical distribution tended to decline in prevalence and contract their range, and maximum temperature tolerance was an important factor in predicting changes in their spatiotemporal trends. These results are consistent with the idea that climate change is affecting the distributions of many fish species along the Texas coast. Increasing diversity is likely a result of northward invasions of tropical species. A previous study in the northeast

United States suggested that the composition of coastal communities might be shifting from fishes to invertebrates[2]. Our results also suggest that a higher proportion of invertebrates increased their occupancy probability than fishes, suggesting that we might also see a gradual shift of species composition from fishes to invertebrates along the Texas coast.

The species that increased in occupancy probability included a mixture of tropical and subtropical species; many are ecologically and economically important in the northern Gulf of Mexico. For example, a variety of coastal and estuarine predators increased in prevalence, such as spotted seatrout (*Cynoscion nebulosus*), red drum (*Sciaenops ocellatus*), black drum (*Pogonias cromis*), and Spanish mackerel (*Scomberomorus maculatus*). In addition to their ecological significance as predators, which can enact top-down control on prey, their populations support extensive recreational fisheries. Similarly, some commercially exploited species that constitute a large portion of the diets of many coastal predators[15] displayed increasing occupancy probabilities, such as Gulf menhaden (*Brevoortia patronus*), Atlantic croaker (*Micropogonias undulatus*), white mullet (*Mugil curema*), brown shrimp (*Farfantepenaeus aztecus*), and blue crab (*Callinectes sapidus*). The increase in prevalence of these critical, fishery-exploited predators and prey will likely result in both ecological and socioeconomic impacts.

The fish species that both increased in occupancy probability and expanded their range included many tropical and structure-associated species, including chain pipefish (*Syngnathus*

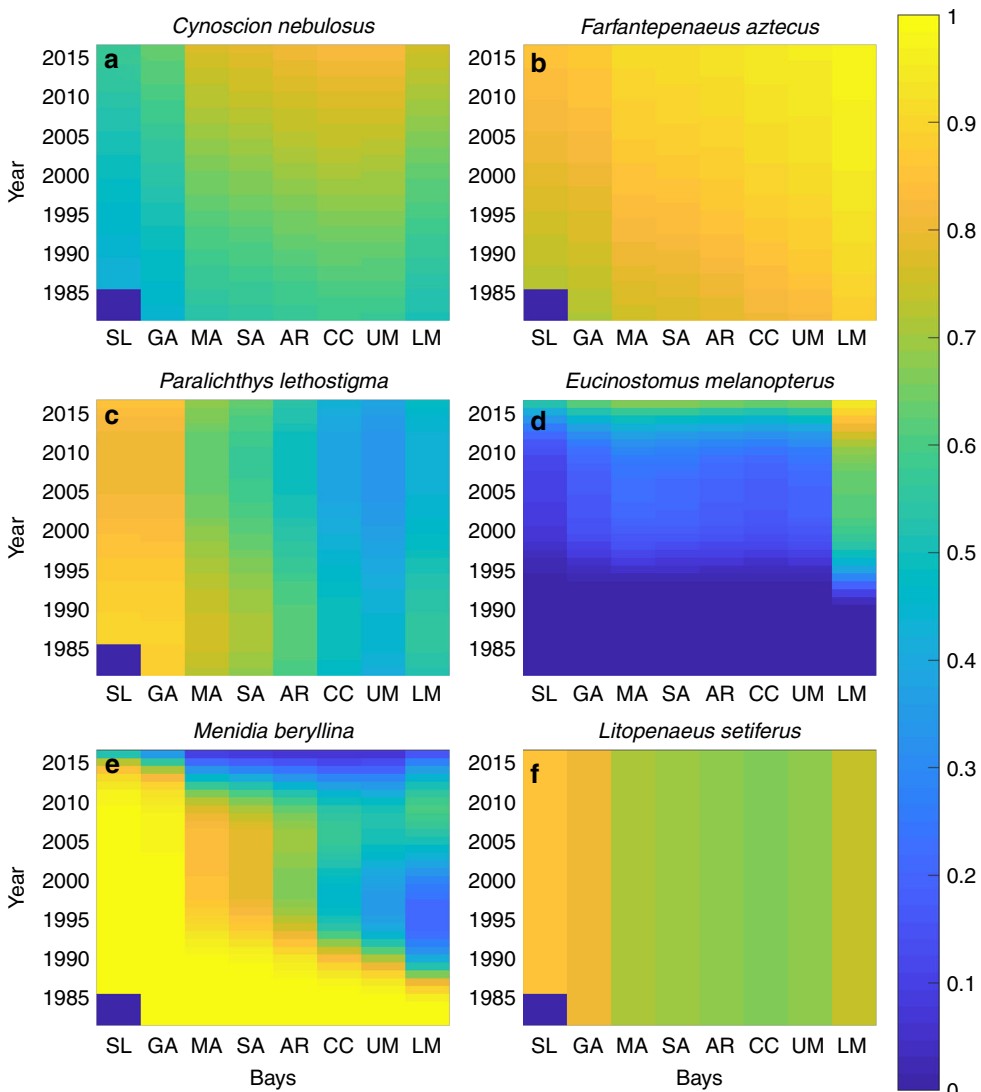

**Fig. 4** Estimated occupancy probability for six species of fishes and invertebrates by using time and latitude as covariates. The occupancy probability for annual mean in a given bay in a given year is plotted. **a** Spotted seatrout (*Cynoscion nebulosus*), increasing over time; **b** brown shrimp (*Farfantepenaeus aztecus*), increasing over time; **c** southern flounder (*Paralichtys lethostigma*), decreasing over time; **d** flagfin mojarra (*Eucinostomus melanopterus*), increasing and expanding over time; **e** inland silverside (*Menidia beryllina*), decreasing and contracting over time; **f** white shrimp (*Litopenaeus setiferus*), no temporal change. The scientific name is shown above each panel. SL Sabine Lake, GA Galveston Bay, MA Matagorda Bay, SA San Antonio Bay, AR Aransas Bay, CC Corpus Christi Bay, UM Upper Laguna Madre, LM Lower Laguna Madre. The results presented in each panel are based on the total of 58,604 measurements made over 35 years in eight bays. See Supplementary Data 4 and 5

louisianae), dwarf seahorse (*Hippocampus zosterae*), smooth puffer (*Lagocephalus laevigatus*), flagfin mojarra (*Eucinostomus melanopterus*), gray snapper (*Lutjanus griseus*), and Gulf kingfish (*Menticcirhus littoralis*). Many of these species rely heavily on seagrasses and submerged vegetation as juveniles and adults, so the expansion of their ranges may suggest changes in the availability of those habitats. The expansion of this assemblage of tropical and subtropical species into the northwestern Gulf of Mexico aligns with previous studies that indicated changes in the structure of coastal communities[16,17]. This northward shift in species' distributions will likely have far-reaching ecological consequences, such as alterations of trophic ecology[18].

Species that exhibited decreasing occupancy probability were primarily subtropical and included some ecologically and economically valuable predator and prey species such as bay anchovy (*Anchoa mitchilli*), hardhead catfish (*Arius felis*), southern flounder (*Paralichthys lethostigma*), Gulf flounder (*Paralichthys*

albiguttata), and crevalle jack (*Caranx hippos*). In this study, marine species that could tolerate freshwater or brackish water (or vice versa) were included in the analysis. Interestingly, many of those freshwater-associated species exhibited decreasing trends in occupancy, including sheepshead minnow (*Cyprinodon variegatus*), longnose killifish (*Fundulus similis*), bayou killifish (*Fundulus pulvereus*), rainwater killifish (*Lucania parva*), and fat sleeper (*Dormitator maculatus*). These species are associated with freshwater outflow and marsh tidal creeks, so the decrease in their occupancy probability may suggest alterations of the availability of those habitats.

Several underlying mechanisms linking climate change and the distribution of marine species have been suggested in recent studies[8–11,19,20]. According to our results, salinity was the main environmental factor in explaining among-bay variability in occupancy probability. This may be related to the strong salinity gradient present in the bays along the Texas coast, which

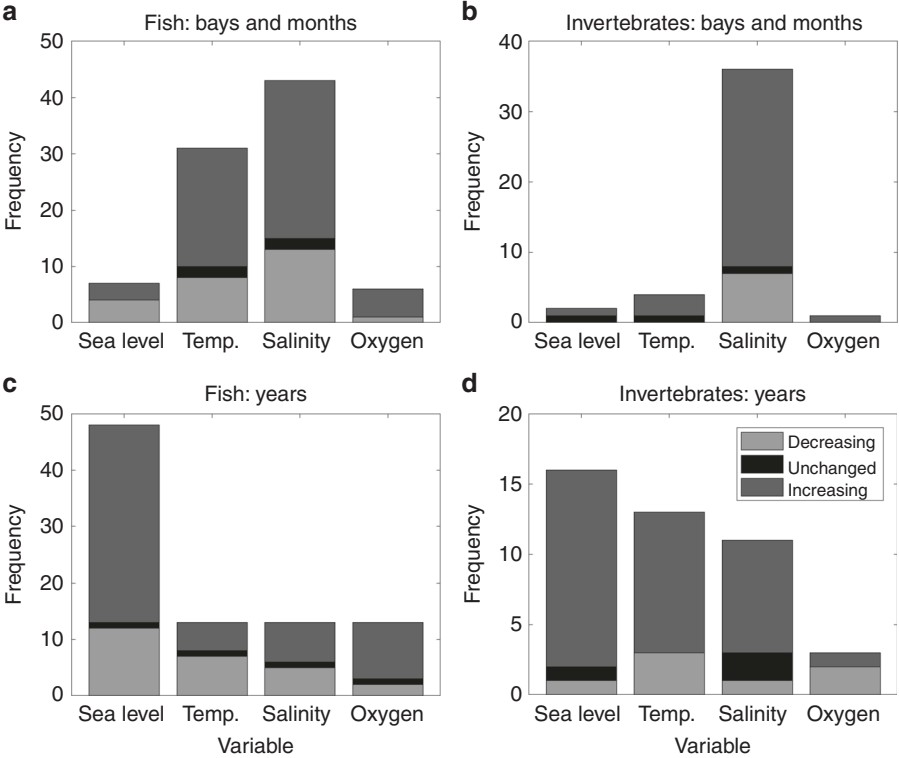

**Fig. 5** Frequency of including specific environmental variables in occupancy probability in the best model. Variable explaining variability among bays and months for **a** fish and **b** invertebrates. Variable explaining variability among years for **c** fish and **d** invertebrates. Species were separated into those decreasing (light shade), increasing (intermediate shade), and unchanged (dark shade) over time. The numbers of samples are 87 fishes and 43 invertebrates. See Supplementary Data 6 and 7

generally increases from north to south. However, climate change is predicted to alter salinity regimes in coastal bays by affecting river discharge, particularly by reducing the availability of surface water in subtropical dry regions[21]. Salinity is known to affect the reproduction, development, and growth of fish[22], and it is considered one of the main factors determining their ontogenetic migration[23]. Our results suggest a need for further investigation of the mechanism behind the effect of salinity on the distribution of coastal species.

Our results also suggest that sea level is another potentially important climate-related factor affecting the distributions of coastal species. Changes in sea level and salinity together can affect the availability of biogenic habitats such as salt marshes, which provide important nursery habitat for many marine organisms[24–26]. Bag seines like those used in this study primarily target small, young-of-the-year, and juvenile fish and invertebrates, which tend to be more strongly associated with those habitats. Therefore, our results are consistent with the notion that changes in the availability of critical structured habitats may be affecting the distribution of coastal marine organisms in our study region. Coupled with our results on salinity, we caution attempts to predict future distributions of marine organisms by solely including the effects of single environmental variables such as temperature.

Although temperature alone may not have been the most important factor affecting the variability in the prevalence of fish and invertebrates in this region, it likely has a synergistic effect with salinity and/or sea level, considering it was the second most important factor explaining variability in occupancy probability (Fig. 5). Along the Texas coast, many species that are adapted to colder, temperate waters are located close to the southern edge of their distribution and likely experience more intense temperature

stress than those adapted to warmer, tropical waters. It is possible that multiple compounding stressors, including altered salinity regimes, changes in habitat availability due to sea-level rise, and increased temperature, may be synergistically driving some species out of the system, and allowing those species that are better adapted to the new conditions to invade. The occurrences of episodic large temperature changes may also affect the distribution of coastal organisms. For example, mass mortality of organisms due to cold temperature was recorded in 1983 and 1989[27], and it is plausible that the reduced occurrences of extremely cold temperature in winter are allowing the invasion of southern species.

Studying the effects of climate change on species distributions of marine organisms is complicated because fisheries can also affect distributions. In our study, the data were collected independently of fisheries (fishery independent); therefore, our analysis is more robust than many previous studies that have used fishery-dependent data. Along the Texas coast, commercial gill nets were banned in 1988, and limited entry programs were implemented for shrimp, crab, and finfish fisheries in 1995, 1998, and 2000, respectively. It may be plausible that the increase in the prevalence of species targeted by commercial fisheries, e.g., brown shrimp, blue crab, black drum, and Atlantic croaker, is due to reduced fishing pressure. However, none of these species are expanding or contracting their range. Furthermore, changes in fishing regulations could have a greater impact on adult populations; however, this study analyzes mostly young-of-the-year data. While stock-recruitment relationships are possible for medium to long-lived species, for short-lived species, such as white shrimp and brown shrimp, this relationship is weak, and recruitment fluctuations have been linked to environmental factors[15]. Therefore, although it remains plausible, changes in fishing

regulations are difficult to explain the trends observed in this study.

Our understanding of the responses of biological systems in the ocean to climate change is lagging far behind investigations of the atmosphere and land[28,29]. Our study demonstrates that the structure of estuarine communities along the Texas coast is already showing signs of such responses. Although some species decreased in their prevalence, diversity is generally increasing, and many species are expanding their range as the availability of suitable environmental conditions shifts northward. This pattern could not have been detected with observations of a small number of species. We suggest continued monitoring studies along the Texas coast and new long-term monitoring programs elsewhere to better understand the responses of biological systems to climate change.

## Methods

**Fish and invertebrate data.** Data were collected under the Marine Resource Monitoring Program conducted by the Coastal Fisheries Division of the Texas Parks and Wildlife Department. In this study, we used data from Sabine Lake, Galveston Bay, Matagorda Bay, San Antonio Bay, Aransas Bay, Corpus Christi Bay, the upper Laguna Madre, and the lower Laguna Madre (Fig. 1, Supplementary Table 1) from January 1982 to December 2016 (except in Sabine Lake, where sampling began in January 1986). The surveys were conducted biweekly by using bag seines (18.3 m long and 1.8 m deep with 19-mm stretched nylon mesh in wings and 13-mm stretched mesh in the bag), which were deployed along the shoreline.

Bag seines were deployed multiple times during the first and second halves of the month in every bay system. We considered each deployment as one sample in this study, and the data were structured by year, month, and bay. Sample size is shown in Supplementary Data 1. The location of each sample was determined by randomly selecting one station from a predefined sampling universe, and once in the field, selecting a section of the available shoreline within that station. At the selected sampling location, the bag seine was extended 12.2 m perpendicularly to the shoreline, then pulled parallel to the shoreline over 15.5 m. The offshore end was then retrieved to the shore while keeping the onshore end stationary and maintaining the full extent (12.2 m) of the bag seine by using a limit line. Further details of sampling protocols are described in the Marine Resource Monitoring Operations Manual[12].

Captured individual fish and invertebrates with length >5 mm were identified to the lowest taxonomic level. For the purpose of the occupancy model analysis, if at least one individual of a species was observed, that species was considered present; otherwise, it was considered absent. The presence–absence of species were determined for each sample.

**Environmental variables.** To describe the variability in occupancy probability, we used monthly mean temperature (Supplementary Fig. 1), salinity (Supplementary Fig. 2), dissolved oxygen (Supplementary Fig. 3), and sea level (Supplementary Fig. 4). The monthly mean sea level was obtained from NOAA Center for Operational Oceanographic Products and Services[30] (Supplementary Table 1). The stations were selected based on the availability of data from January 1982 to December 2016. During each bag seine sampling event, temperature, salinity, and dissolved oxygen were recorded. To obtain the monthly mean values, we used the average values for each bay by month. For each river, the station closest to the mouth of the river was selected among those that consisted of the data from January 1982 to December 2016. For the bays without major rivers, the closest river for which data were available was matched. All of the environmental variables were standardized by taking the z-score.

**Rarefaction analysis.** Rarefaction analysis[13] allows for the estimation of species diversity (i.e., species richness, Shannon diversity index, Simpson diversity, or other diversity metrics) that one would expect to observe, given a particular level of sampling effort, for a given site. In general, as the effort increases, the number of species observed increases. Thus, when comparing diversity across multiple sites, it becomes necessary to account for the level of effort used at each site. In order to account for the variability in sampling effort, the rarefaction analysis uses interpolation to estimate the curve for the number of species (or other diversity metrics) one would expect to encounter as a function of the number of samples taken or number of individuals sampled (individual-based vs. sample-based methods). The resultant curve extends from an effort of 0 to the true effort for each site, thereby allowing for all sites to be compared at the same level of effort (i.e., the true effort for the least sampled site). Chao et al.[31] extended this method (as applied to Hill numbers) to include not only interpolated values, but also extrapolated values, where the asymptote of this curve is assumed to be an estimate of the true diversity metric given infinite sampling effort.

For this study, we used the sample-based rarefaction analysis method in which each deployment of a bag seine was treated as one sample, and the presence/absence of each species was determined, in order to estimate the Shannon diversity index for each bay in each year. Samples from each bay and each year were pooled, and the data were randomly sampled many times at each level of effort (i.e., from 1 to n where n is the number of samples for a given bay in a given year) in order to estimate the expected diversity given a particular level of sampling effort. By using the method of Chao et al.[31], rarefaction curves were extrapolated out to their asymptotes in order to estimate the asymptotic Shannon diversity index in a given year for each bay, which we consider to be an estimate of the true diversity of the bay. We only included species that were found at least twice in the data to eliminate outliers. The construction of the rarefaction curves and extrapolation of the asymptotic values were done in the R statistical environment by using the package iNEXT[32]. In addition to calculating the asymptotic Shannon diversity index, the program also calculated the standard error for each estimate following the method described in Chao et al.[31].

Once the diversity index was estimated, it was simulated for each bay of each year by assuming a normal distribution. Then, linear regression was applied to the simulated index against year, noting if the slope was significantly positive, significantly negative, or not (two-tail t test, $\alpha = 0.05$), and conducted 1000 iterations of this process. Because the significance level was merely used as a threshold for the bootstrap analysis, it was not adjusted for multiple comparisons. If the slope was positive (or negative) for a significant number of times ($\alpha = 0.10$), the Shannon index was considered to have significantly increased (or decreased) over time for that bay. This analysis was done for each bay for fish and invertebrates separately. Because eight tests were conducted to test the hypothesis that each taxa was significantly increasing, the p-values were adjusted based on the Holm–Bonferroni[33,34] method with eight comparisons.

**Occupancy model.** The occupancy analysis[14] allows the estimation of probability of occupancy $\psi_{s,y,m,b}$ of species $s$ at location (bay) $b$ in month $m$ of year $y$. For occupancy analysis, we considered each bay in each month in each year as one sample, and multiple deployments of bag seine in each sample as repeated measurements. For this analysis, we only included fish and invertebrate species that were present in more than 30 measurements (out of 58,604 measurements combining all samples). This threshold was arbitrarily chosen to eliminate species for which occupancy probability was clearly inestimable. Then, if the species was not present in all bays in all years in a given month, that month was eliminated from further analysis for that species because occupancy and detection probabilities are confounded and cannot be estimated separately. During those months, they were most likely unavailable (zero occupancy probability) because of their seasonal migration or ontogenetic development (e.g., too small or too large to be sampled in the month). Elimination of the month should not affect occupancy probability estimates for other months. We also eliminated species that were considered strictly freshwater species (i.e., not cross-listed as marine and/or brackish species) according to FishBase[35] as those fishes are directly affected more by river conditions.

The occupancy probability was estimated accounting for the fact that not all species were captured even if they were available to be potentially captured in a sampled area. The probability of detection of a species is denoted by $p_s$ for species $s$, which measures the probability of capturing (i.e., detecting) the species, given that the species is available in the sampled area. By using the two probabilities $\psi_{s,y,m,b}$ and $p_s$, likelihood functions associated with the presence/absence data were expressed for each species separately to estimate the associated parameters.

The detection probability may also vary among locations and time. In our analysis, we used two models for describing $p_s$. In the first model, we assumed $p_s$ to be distributed according to the β-distribution; this is generally known as a random effect model. In the second model, we assumed it was constant, and we call this model a fixed-effect model.

It was impossible for most species to estimate a separate occupancy probability for each bay in each month of each year. Therefore, the occupancy probability was modeled as a function of covariates. There are two separate purposes for the analysis. First, to visualize the occupancy probability and investigate spatiotemporal variation, it was smoothed by using year $Y$ ($Y$: 1982–2016) and latitude $L_b$ (after scaling both by taking the z-score) as covariates and month ($m$: 1–12) as a fixed factor

$$\text{logit}\left(\psi_{s,y,m,b}\right) = \alpha_s + \sum_{k=1}^{d_1} \beta_{s,k} Y_y^k + \sum_{k=1}^{d_2} \chi_{s,k} L_b^k + \delta Y_y L_b + \gamma_{s,m}, \qquad (1)$$

where $\alpha_s$, $\beta_{s,k}$, $\delta$, and $\gamma_{s,m}$ are the coefficients of the linear equation to be estimated. The orders of polynomials were determined by $d_1$ and $d_2$ for the year and latitude effects, respectively. The month effects $\gamma_{s,m}$ were not estimated for the months that were eliminated from the analysis (see the description in *Fish and Invertebrate Data*), and one of the remaining months was selected as a reference level (i.e., factor $\gamma_{s,m}$ was set to 0).

A total of 64 models were constructed for each species by varying $d_1$ and $d_2$ from 1 to 3, by including or excluding the year effect, latitude effect, and the interaction term, and by using either the random or fixed-effect model for the detection probability. Then the best model was selected by using Akaike information criteria (AIC) for each species. The results are shown in Supplementary Data 4 and 5.

We next determined the influence of environmental conditions (sea level, temperature, salinity, and dissolved oxygen) on the occupancy probability. For this purpose, we fitted the following occupancy probability model to the data

$$\mathrm{logit}\left(\psi_{s,y,m,b}\right) = \alpha_s + \sum_{k=1}^{d_3} \mu_{s,1,k} C_{m,b}^k + \sum_{k=1}^{d_4} \mu_{s,2,k} C_y^k \qquad (2)$$

where $C_{m,b}$ is the environmental covariate that was averaged over years and $C_y$ is the environmental covariate averaged over months and bays. The former covariate explained the variability among months and bays, and the latter covariate explained the variability among years. The parameters in the occupancy probability model were $\alpha_s$, $\mu_{s,1,k}$, and $\mu_{s,2,k}$. By varying $d_3$ and $d_4$ from 1 to 3 or by excluding one or both of the covariates and by using fixed or random effect model for detection probability, we developed 32 different models for each species. Then, the best model was selected by using AIC. The covariates included in the model with the smallest AIC were considered the best covariates.

The likelihood was maximized to estimate the parameters by minimizing the negative natural log of the likelihood by using a built-in optimization function (fminunc.m) in MATLAB[36]. Each minimization was repeated ten times with different randomly selected initial conditions. The parameters were considered estimable if the estimated Hessian matrix was invertible (i.e., covariance matrix was estimable). If not all parameters were estimable in all ten replicates, the model for that species was eliminated from further analysis.

**Determination of trends.** Once we estimated the occupancy probability $\psi_{s,b,t}$, we determined the existence of temporal trends based on inclusion of coefficient(s) on the year covariate. If the year covariate was not included in the best model (i.e., all of the coefficients were 0), we considered that the occupancy probability remained constant over time. If at least one of the coefficients was included, we further compared the estimated mean probability in year 1982 and that in year 2016 of the best model, where the mean is taken over bays and months. If the former is larger than the latter, we considered that the occupancy probability had increased over time, and if the former is smaller than the latter, we considered that the occupancy probably had decreased over time. This is equivalent to comparing the sum of all annual changes in the occupancy probability and asking if the sum is >0 (i.e., occupancy probability has increased) or <0 (i.e., occupancy probability has decreased).

For determining range expansion, we took the estimated occupancy probability in year 1982 and 2016 from all of the bays excluding Sabine Lake. Then, we calculated the number of bays with occupancy probability >0.1, which was an arbitrarily determined threshold based on visual inspections of figures. If the number of the bays increased from 1982 to 2016, we considered that there was range expansion. If the number decreased, we considered that there was range contraction. Otherwise, we assumed that there was no range expansion or contraction.

**Distribution and temperature range.** The distribution and temperature range of fish species were obtained from FishBase[35] by searching the combination of genus and species name. For each species, latitudes for northern and southern edges of their distribution were recorded. When the southern edge was in the southern hemisphere, we recorded 0°N for the southern edge as we were only interested in the distribution in the northern hemisphere. For temperature range, FishBase includes an estimated preferred temperature range for the majority of the species that we investigated. If the estimated preferred temperature range was not available but the observed temperature range was noted, we used the observed range for that species. For the analysis, the maximum preferred/observed temperature was used.

The four variables (maximum preferred/observed temperature, southern distribution limit, northern distribution limit, and range) were compared between species that increased and decreased occupancy probabilities and between species that contracted and expanded their range. Since four tests were conducted for each of the two comparisons, we adjusted the p-value by using the Holm–Bonferroni method[33] with four comparisons.

**Statistics and reproducibility.** Details of all data and statistical analyses are presented in the "Methods" section for reproducibility. Custom code for occupancy analysis is available from GitHub[37], and the raw data are available from BCO-DMO[38].

**Reporting summary.** Further information on research design is available in the Nature Research Reporting Summary linked to this article.

## Data availability

Data used in this analysis are available from BCO-DMO[38].

## Code availability

Rarefaction analysis was done with R 3.5.2 with the package iNEXT. Occupancy analysis was done with MATLAB with a built-in optimization function (fminunc.m). Custom codes for likelihood are available from GitHub[37].

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

## Acknowledgements

This project was funded by NSF OCE 1656923 to MF. We thank Jasmin Diaz-Lopez for preparing the map of sampling locations.

## Author contributions

M. Fujiwara, F.M.A., and M. Fisher conceived of the project idea. F.M.A. and M. Fisher provided the fish and invertebrate data. M. Fujiwara conducted the occupancy analysis, and M.P. conducted the rarefaction analysis. M. Fujiwara, F.M.A., R.J.D.W., M. Fisher, M.P., and M.C.L contributed in the interpretations of the results and writing of the paper.

## Competing interests

The authors declare no competing interests.
