## [Peer Review File · Communications Biology]

Reviewers' comments:

Reviewer #1 (Remarks to the Author):

I have read the paper "Climate-Related Changes in the Diversity of Fish and Invertebrates in Subtropical Bays" by Fujiwara et al. concerning change in fish community structure in bays and estuaries in the Gulf of Mexico. The paper describes changes in diversity and occupancy among these systems, putative association with climate change were made. More specifically, they lay out the case that certain variables associated with the chemistry of these systems seem most important. The systems have experience shifts in salinity gradients over time, suggesting associated factors such as nursery habitats have changes, which may be more important than change thermal regimes. The main concern I have about this paper is that it is not clear whether we are also seeing a response to change fishery pressure. Near the end of the paper, fishing is discussed; however, it is not addressed as a factor in allowing for the expansion of species distributions. I believe the authors need to address this issue in more detail. The paper was well written and logical in its presentation of the evidence. The only recommendation I have on the text is that the first few line of the main text contain some awkward phases and it lacks the impact that I think the paper deserves.

Reviewer #2 (Remarks to the Author):

1. Brief summary of the manuscript

This paper looked at regional specific (nine bays along the Texas Coast, Gulf of Mexico) changes in diversity, probability of occupancy, and descriptors of those changes. The authors found that in general, diversity is increasing and many species are expanding their range. They also found that species with declining occupancy probability were at the northern edge of their range and lower max/preferred temperature. Species that were contracting their range were at the southern edge of their range.

2. Overall impression of the work

Overall, this work was very interesting and contributes to the pressing need to understand how different systems and different species are responding to climate change and all of its various emergent properties. I think with a little bit more information and some statistical adjustments, this will make a great contribution to regional knowledge. The methods that require more detail and/or adjustments are listed below in overall impressions and specific comments.

Statistically, it does not appear that any of the analyses were Bonferroni corrected. There are a lot of statistical significance comparisons made from the same data set using many different approaches, and so I fear that some of the results are just an artifact of the many comparisons. I suggest looking into adjusting p-values for each of these statistical tests given multiple comparisons (e.g. significance of Shannon index).

I would like to see AIC and variance explained reported for some of the top models so that the reader can determine if there is a difference among models tested. I would also like to see more explanation surrounding the removal of covariates while testing for significance – the order of removal of covariates can influence the significance, especially in this case where different combinations of covariates were used to predict the same thing (detection probability) in multiple models.

How were preferred temperatures used in describing the occupancy probability trends determined for each species?

Did you test for variance inflation factor (VIF) to determine issues of collinearity in the covariates used to describe ocean/environmental characteristics?

It was a little hard to keep track of all the analyses/comparisons that were done and the results. Is it possible to add a simple table with an overview of analyses and results?

3. Specific comments, with recommendations for addressing each comment

Line 33: "impact the socioeconomics of people" awkward phrasing.

Line 91 – 92: "result in both ecological and socioeconomic impacts" such as?

Lines 272 – 275: Why did you use both random and fixed effects for detection probability? Did you use model selection to choose between these two? Shouldn't the detection probability be fixed (since that's what you're investigating) and then some of the covariates to describe this probability can be random?

Lines 279 – 280: Year and month were factor variables to prevent influence of numerical value on the ranking/estimates/relative effect of these covariates, correct?

Line 287: Change "The total" to "A total"

Line 287: Again, 64 models per species (94 fish, 45 invertebrates) means there are a lot of models. If you're just looking at Information Criterion, ok. But again, it seems like you're drawing information on the significance of covariates with likely thousands of comparisons. Adjust p-values appropriately.

Line 276 – 301: Why do you have separate occupancy probability models? Can't all covariates be combined into one single model and tested at once? That way, the variance explained can contain all covariate information.

Line 316 – 318: How did you decide if that increasing or decreasing trend was beyond the long-term variability, though?

Line 321: Describe why you picked 0.1. Arbitrary decisions are made often, just need to justify this value.

Line 322 – 324: Again, just an increase or decrease in mean value doesn't suggest to me that there is a change in range expansion/contraction. How do you determine that this is different than the data noise/long-term variability?

Line 340: Change "Map showing bays where samples were taken." To "Location of bays used in study."

Line 346: I would be interested in all p-values and sample sizes for all locations.

Line 362: I really like this plot!

Line 377: Figure 5 would be easier to interpret if you just change the x-axis labels to the full variable

name. Also am I correct as interpreting this as salinity and sea level explained the majority of variability in detection probability, but that temperature and dissolved oxygen still explained some of the variance? This seems worthwhile noting in the discussion, especially where you discuss each specific type/group of fish/invertebrate response.

Reviewer #1 (Remarks to the Author):

I have read the paper “Climate-Related Changes in the Diversity of Fish and Invertebrates in Subtropical Bays” by Fujiwara et al. concerning change in fish community structure in bays and estuaries in the Gulf of Mexico. The paper describes changes in diversity and occupancy among these systems, putative association with climate change were made. More specifically, they lay out the case that certain variables associated with the chemistry of these systems seem most important. The systems have experience shifts in salinity gradients over time, suggesting associated factors such as nursery habitats have changes, which may be more important than change thermal regimes. The main concern I have about this paper is that it is not clear whether we are also seeing a response to change fishery pressure. Near the end of the paper, fishing is discussed; however, it is not addressed as a factor in allowing for the expansion of species distributions. I believe the authors need to address this issue in more detail. The paper was well written and logical in its presentation of the evidence. The only recommendation I have on the text is that the first few line of the main text contain some awkward phases and it lacks the impact that I think the paper deserves.

Responses: Discussion included one paragraph on the effects of reduced fishing pressure potentially contributing to the expansion of species. This paragraph was further expanded for clarification. The new paragraph explicitly points out that the species targeted by commercial fishing did not expand. Thus, the reduced fishing pressure on these species cannot explain why other species expanded. We further include the sentence “Therefore, although it remains plausible, changes in fishing regulations are difficult to explain the trends observed in this study.”

The first paragraph of the manuscript was re-written. This should have eliminated some awkward phrases.

Reviewer #2 (Remarks to the Author):

1. Brief summary of the manuscript

This paper looked at regional specific (nine bays along the Texas Coast, Gulf of Mexico) changes in diversity, probability of occupancy, and descriptors of those changes. The authors found that in general, diversity is increasing and many species are expanding their range. They also found that species with declining occupancy probability were at the northern edge of their range and lower max/preferred temperature. Species that were contracting their range were at the southern edge of their range.

2. Overall impression of the work

Overall, this work was very interesting and contributes to the pressing need to understand how different systems and different species are responding to climate change and all of its various emergent properties. I think with a little bit more information and some statistical adjustments, this will make a great contribution to regional knowledge. The methods that require more detail and/or adjustments are listed below in overall impressions and specific comments.

Statistically, it does not appear that any of the analyses were Bonferroni corrected. There are a lot of

statistical significance comparisons made from the same data set using many different approaches, and so I fear that some of the results are just an artifact of the many comparisons. I suggest looking into adjusting p-values for each of these statistical tests given multiple comparisons (e.g. significance of Shannon index).

Response: The Bonferroni correction corrects the significance level rather than p-value. Under the Shannon index comparisons, all of the p-values are almost 0. Therefore, there is no change in the conclusion. Because we show the p-values, this should be clear to the readers. We also show the p-values for those that are not significant in the revised version.

I would like to see AIC and variance explained reported for some of the top models so that the reader can determine if there is a difference among models tested. I would also like to see more explanation surrounding the removal of covariates while testing for significance – the order of removal of covariates can influence the significance, especially in this case where different combinations of covariates were used to predict the same thing (detection probability) in multiple models.

Response: The important patterns in this study emerge from investigating many species simultaneously rather than examining each species separately. Although there is indeed model uncertainty with each species, the results from our analysis with many species should be robust against the model uncertainty. There are approximately 150 top models. If we include three top models for each, there will be 450 top models. It would be too much to include.

The order of removal will not influence the results because we examined all possible combinations of covariates and selected the best model for each species. The detection probably does not use any of the environmental covariates. This is described in the method.

How were preferred temperatures used in describing the occupancy probability trends determined for each species?

Response: Preferred temperature was compared between those increasing in occupancy probability and others decreasing in occupancy probability using Mann-Whitney U test).

Did you test for variance inflation factor (VIF) to determine issues of collinearity in the covariates used to describe ocean/environmental characteristics?

Response: No, because collinearity is avoided in the analysis. First, river discharge was eliminated because it is strongly correlated with salinity. Then, at most two variables were included in each model. But for one of them, mean over years was taken, and for the other, mean over space was taken. Therefore, there is no correlation between the covariates in each model.

It was a little hard to keep track of all the analyses/comparisons that were done and the results. Is it possible to add a simple table with an overview of analyses and results?

Response: After substantial revision, we believe it is much easier to follow the analyses. We conducted four major analyses, and the result section includes four paragraphs, each corresponding to one of the four major analysis.

3. Specific comments, with recommendations for addressing each comment

Line 33: “impact the socioeconomics of people” awkward phrasing.

Response: This phrase was deleted.

Line 91 – 92: “result in both ecological and socioeconomic impacts” such as?

Response: This phrase was deleted.

Lines 272 – 275: Why did you use both random and fixed effects for detection probability? Did you use model selection to choose between these two? Shouldn't the detection probability be fixed (since that's what you're investigating) and then some of the covariates to describe this probability can be random?

Response: There were two types of models (random effect and fixed effect models) for detection probability. Because detection probability may vary substantially among sampling, we used random effect model. The model selection was used to select the best model. We did not use any covariate for detection probability because the environmental variables probably affected occupancy rather than detection probability.

Lines 279 – 280: Year and month were factor variables to prevent influence of numerical value on the ranking/estimates/relative effect of these covariates, correct?

Response: There are 12 levels for month. In other words, each month has a different intercept. This is to account for seasonality. Year in the first occupancy analysis was covariate to capture temporal trends. Then, when environmental variables were included, variation in occupancy probability among years came from variation in the environmental variable.

Line 287: Change “The total” to “A total”

Response: Changed as suggested.

Line 287: Again, 64 models per species (94 fish, 45 invertebrates) means there are a lot of models. If you're just looking at Information Criterion, ok. But again, it seems like you're drawing information on the significance of covariates with likely thousands of comparisons. Adjust p-values appropriately.

Response: We are only looking at AIC. All of the tests are based on the patterns (increasing or decreasing) in the results that were selected under AIC. We are dealing with a large number of species, and we are seeking the significant patterns in the results using statistical tests.

Line 276 – 301: Why do you have separate occupancy probability models? Can't all covariates be combined into one single model and tested at once? That way, the variance explained can contain all covariate information.

Response: This comment contradicts with the previous comment. This is suggesting a large number of significance test. Furthermore, this comment contradicts with the earlier comment about collinearity. It is not ideal to include many covariates at once and test their significance.

Line 316 – 318: How did you decide if that increasing or decreasing trend was beyond the long-term variability, though?

Response: We did not. We simply determined whether the occupancy probably has increased overall from 1982 to 2016. It is possible that some of them exhibited periodic patterns, for example, and 1982 happened to be trough year and 2016 happened to be ridge year within the periodicity. This type of problems is common problems in climate change research. In our study, however, we conducted further analyses to determine whether we can explain the apparent increasing trend and decreasing trends with the characteristics of species that are exhibiting these patterns. In addition, we also associated the occupancy probability with environmental variables. Sea level, which is clearly increasing over time, explained the temporal change in the occupancy probability. Of course, the change in the sea level itself may be a part of a periodic pattern, but it is a beyond the scope of this study to determine the cause of sea level rise.

Line 321: Describe why you picked 0.1. Arbitrary decisions are made often, just need to justify this value.

Response: It now states “based on visual inspection of figures.” However, it is arbitrary determined, and if we can justify it, it is no longer arbitrary decision. We just needed to pick a small number greater than 0.

Line 322 – 324: Again, just an increase or decrease in mean value doesn’t suggest to me that there is a change in range expansion/contraction. How do you determine that this is different than the data noise/long-term variability?

Response: We are not sure what types of data noise the reviewer is referring. We did occupancy probability analysis to account for variation in detection probability, which accounts for sampling variation. There could be spatial heterogeneity of organisms; it could lead to detection probability. We try to reduce this type of noise by obtaining samples from multiple locations within each bay in each month. There are many different types of noises, and we think we eliminated them as much as possible based on the available data, which are very extensive data sets compared with many other ecological studies. As for “long-term variability”, we did not eliminate the possibility as we replied previously.

Line 340: Change “Map showing bays where samples were taken.” To “Location of bays used in study.”

Response: This change was made.

Line 346: I would be interested in all p-values and sample sizes for all locations.

Response: The p-values are shown.

Line 362: I really like this plot!

Response: We also think the figures (including the ones in Supporting Information) are very informative.

Line 377: Figure 5 would be easier to interpret if you just change the x-axis labels to the full variable name. Also am I correct as interpreting this as salinity and sea level explained the majority of variability in detection probability, but that temperature and dissolved oxygen still explained some of the

variance? This seems worthwhile noting in the discussion, especially where you discuss each specific type/group of fish/invertebrate response.

Response: The labels were changed. The interpretation is not correct. All of the covariates are for occupancy probability. To clarify this the figure legend explicitly states “in occupancy probability.”

Reviewers' comments:

Reviewer #1 (Remarks to the Author):

I continue to think that this paper will be of wide interest and feel the changes made by the authors were positive in improving the paper. I can recommend the paper for publication.

Reviewer #2 (Remarks to the Author):

To follow best statistical practices, I still believe authors need to adjust their overall significance levels (using an established method such as Bonferroni correction) in many of their statistical tests that are relying on alpha to justify study claims. Authors may find their significance level (alpha) required severely reduced based on thousands of comparisons from the (same?) data set if adjusted. As well, if their p-value results are exactly zero, they may want to increase the significant figures in their decimals as this seems unlikely. It still isn't clear to me though how many comparisons were done with one data set. These adjustments will likely influence some of their results, or better justify that their hundreds/thousands of comparisons aren't a result of spurious positives.

e.g. with p-values close to 0.05, Mann-Whitney results, others such as "Our results showed 86 that the proportion of declining fish species (26 out of 87 species) was significantly greater than that of 87 declining invertebrate species (7 out of 43 species; Single-Tail Binomial Test, p-value=0.041)."

I also think the fishing effort is a huge variable to leave out of a spatiotemporal analysis of fish in Gulf. I'm not sure how they are concluding in the discussion that "changes in these fishing regulations cannot explain the temporal association between the prevalence of a large number of species" (lines 192 - 93)... was this tested? Even non-target species are still impacted by fishing (bycatch, predator-prey dynamics, habitat competition) and fishing effects are often lagged, especially if using fishing regulation as an indicator of fishing effects/effort on the ecological community.

Other than that, most comments were addressed.

We would like to thank the two reviewers for taking valuable time providing comments. There were two points made: adjustment of p-value and effect of fisheries. We revised the manuscript based on these comments.

As for the adjustment of p-value, we adjusted it using the Holm-Bonferroni method. At the significance level of 0.1, none of the conclusions has changed. The adjustment of p-value is tricky (we are not sure if it is “the best statistical practices”). For example, when northern range limit was included in the analysis, we expected insignificant result. Ensuring a negative result is often very informative, but although it should not influence other significant results, the adjusted p-values are affected in the current calculation done as suggested by reviewer 2. It is important to think why adjustment is needed. Provided the comments are made available to the readers, we are fine either way (using adjusted p-values or original p-values) because the conclusions do not change. This is an editorial decision because sufficient information is included in the manuscript now.

As for 1,000 tests that the reviewer is referring, they are not used as tests. They are merely the criteria to include or exclude in a bootstrap method. This is clarified in the method section. In most of the significant cases, the slope is positive (according to the criteria) almost 1,000 times out of 1,000 simulations. They are significant and visually obvious. We would be happy to conduct other statistical tests if alternative tests are suggested. Either way, we think the results are very clear.

As for the comment on fishery, the discussion is our interpretations of the results. We demonstrated the associations between change in the prevalence of species and environmental variables. We suggested that these relationships cannot be explained by fishing effort. Obviously, fishing should not affect the environmental variables. If fishing is affecting the prevalence, then both fishing and climate change may be affecting fish and invertebrate prevalence. Therefore, we do not exclude the possibility of fishing effort affecting some organisms as this has been clearly stated in the paragraph. The comment is understandable because fishery biologists often think fishing is the main (sometimes only) factor regulating marine organisms. Our results indicate physical environmental conditions can explain the observed patterns in prevalence of fishes and invertebrates. This makes our results more interesting to the readers.